# Hash-Based Deep Learning Approach for Remote Sensing Satellite Imagery Detection

**Samhitha Gadamsetty [1], Rupa Ch [1,\*], Anusha Ch [1], Celestine Iwendi [2] and Thippa Reddy Gadekallu [3,\*]**

1   Department of Computer Science, VR Siddhartha Engineering College, Vijayawada 520007, India; samhitha126@gmail.com (S.G.); anusha3619@gmail.com (A.C.)
2   School of Creative Technologies, University of Bolton, Bolton BL3 5AB, UK; celestine.iwendi@ieee.org
3   School of Information Technology and Engineering, Vellore Institute of Technology, Vellore 632014, India
\*   Correspondence: rupamtech@gmail.com (R.C.); thippareddy.g@vit.ac.in (T.R.G.)

**Abstract:** Ship detection plays a crucial role in marine security in remote sensing imagery. This paper discusses about a deep learning approach to detect the ships from satellite imagery. The model developed in this work achieves integrity by the inclusion of hashing. This model employs a supervised image classification technique to classify images, followed by object detection using You Only Look Once version 3 (YOLOv3) to extract features from deep CNN. Semantic segmentation and image segmentation is done to identify object category of each pixel using class labels. Then, the concept of hashing using SHA-256 is applied in conjunction with the ship count and location of bounding box in satellite image. The proposed model is tested on a Kaggle Ships dataset, which consists of 231,722 images. A total of 70% of this data is used for training, and the 30% is used for testing. To add security to images with detected ships, the model is enhanced by hashing using SHA-256 algorithm. Using SHA-256, which is a one-way hash, the data are split up into blocks of 64 bytes. The input data to the hash function are both the ship count and bounding box location. The proposed model achieves integrity by using SHA-256. This model allows secure transmission of highly confidential images that are tamper-proof.

**Keywords:** remote sensing; deep learning; ship detection; YOLO v3; hash value

## 1. Introduction

In contrary to machines, it is easy for humans to detect, classify, and identify objects that are present in their surroundings irrespective of how they are positioned, aligned, etc. These objects can be easily identified even if they are misplaced, or they have different visuals. This signifies that object detection in human sense is frivolous. If the same process is to be carried out through a machine, a great deal of computational work and energy is required to process the information related to the object and to identify to which category the object belongs to. Machines can handle this sort of object detection by using either an image input or a video input. There are many stages involved in object detection. They can be specified starting from extracting the features of the object and then processing those features, followed then by object classification. Each of the stages involved in object detection consists of many strategies that provide different performances in different circumstances.

The applications of object detection can be found in various territories, such as image retrieval, object tracking, computerized vehicle systems, machine investigation, and many more. However, still, there remain several difficulties with respect to object detection, as there are requirements that are inestimable in consideration to the applications of object detection. On the other side, satellite images offer compelling information. They usually are made up of a large number of pixels in which their size range varies from tens of centimeters to tens of meters. Satellite images can be chosen from any category, such as visible imagery, infrared imagery, and water vapor imagery based on the requirement.

Ship detection using satellite images is one the challenging applications in object detection because it requires high accuracy of the result, as it deals with national security and maritime surveillance. Many strategies have been proposed for object detection, but they face problems with respect to accuracy and efficiency. Thus, in this study, a new methodology is proposed for ship detection using satellite images that are embedded with security. The model that is proposed has advanced efficiency and accuracy for the following reasons:

(i)　YOLOv3 performs object detection at three different levels for different sizes of objects, which boosts the performance.

(ii)　The model is concatenated with extra security using hashing so that potential attackers find it difficult when information related to such national security is transmitted.

The remaining paper is organized as follows: Section 2 represents literature survey followed by the proposed methodology, then followed by result and analysis, and is concluded in conclusion section. Figure 1 represents the generic diagram for the YOLOv3. The diagram below depicts how detection is done for different object sizes at different levels. At level 82, tiny objects are detected. At level 94, medium-sized objects are detected. At level 106, large objects are detected.

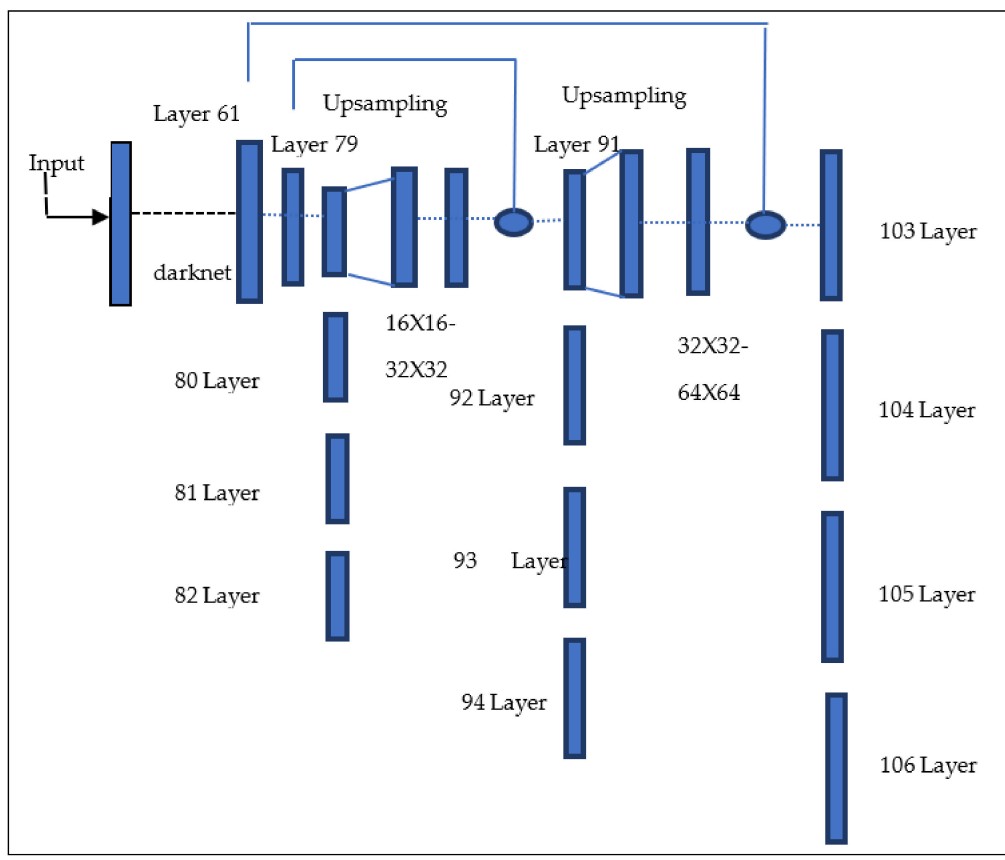

**Figure 1.** Generic Diagram.

## 2. Materials and Methods

The model is fed using Kaggle "air bus ship detection" dataset. This dataset consists of 231,722 images. The model is trained with 70% of the data and tested with 30% of the data for cogency. The dataset consists of only two attributes, namely image ID and encoded pixels, which are used for the ship detection and visualization purpose. The dataset consists of distinct images as a result of different climatic conditions ranging from land masking, cloud masking, and also fog satellite images. Effective data pre-processing techniques are enforced for ensuring the accuracy and effectiveness of the model. The



model is executed using Jupyter Notebook platform. The programming language used for wavering the model is python. The following research manuscripts were referred as a part of literature survey.

Xi Li et al. [1] discussed a model for ship detection and classification using k-Means clustering algorithm. As the input images were very large, it cuts the large image into very small images using window-sliding technology. It can detect only a single kind of ship. To improve the accuracy during detection, it uses k-Means and then reclusters the anchor box. A method called multi-level feature extraction is used to detect even a small target, as the algorithm divides it into smaller images. Hence, it improves detection accuracy for smaller images, too. The main features are as follows: (i) Recognition speed is very high. (ii) Missing ship detection and false detection perform well even in cloud cover situations. The main pitfalls in this analysis are the following: (i) It cannot accurately detect ships between sea and land interference region. (ii) If the input size is too small, further reduction based on window-sliding technology may increase the complexity, as detection must take place at more minute level.

Hao Li et al. [2] developed a model based on SDVI, as this provides the best accuracy compared to other models. SDSOI images were used for ship detection in this paper. Enhanced YOLOV3 real-time network is used for tiny particles detection. It is 9.6% more efficient than SOAT. The main pitfalls in this analysis are that (i) variation in sizes of the input affects the performance, and (ii) it demonstrates poor tolerance to noise.

Wang Yu et al. [3] developed a model to detect the images of ship in high-resolution remote sensing images. SDT filter is used to perform the pre-processing, which reduces the noise. Then cloud masking and land masking are thus completed. SVM, RFA, and SDA machine learning models were used for detection. The main limitations in this work are that (i) complete cloud masking was not done, and (ii) while filtering the image, the target may sometimes be removed.

Zhonghua Hong et al. [4] designed a model that mainly concentrates on detection in complex sea surfaces. SAR and optical imagery was used to overcome this. This paper used liner scaling by means of k-Mean++ algorithm. Uncertainty estimators were used for the positioning of bounding boxes. Finally, four anchor boxes were used to detect the recovery from it. The main drawback in this work are as follows: (i) It can only be 100% efficient in high-resolution optical images; (ii) cloud and cloud shadow may affect the ship detection.

Bo Li et al. [5] collected methods for various detections and classifications for detection in satellite images. This paper discusses about various feature extraction strategies. Then public datasets were applied on the models to check their efficiency. The limitation in this work is that the environmental factors greatly influence the accuracy in the detection of ship. Xuan Nie et al. [6] proposed a model that performs segmentation to the pixel level, as all the existing models only cover the bounding box through the ground level. This paper proposes an approach to improve the R-CNN approach. By following this approach, it can detect and segment ships at the pixel level. It adds the bottom-up structure to the Mask R-CNN model, which improves accuracy. The main limitations include the loss of degree of information in small targets.

Guang Yang et al. [7] proposed a model that can detect ships automatically in high-resolution images. It proposes a sea surface analysis model to solve this problem. It checks whether the sea surface is homogenous. False alarms are removed using the length-width ratio. It automatically gives weights to function, called candidate selection. The main limitation is that it does not show the yield results during cloud coverage and strong wave conditions. Richard Olsen et al. [8] developed a model that concentrates on High-Resolution Synthetic Aperture Radar images. Land-based and space-based data were identified to detect vessels in High-Resolution Synthetic Aperture Radar images. It shows results on high-resolution RADARSAT-2 images. The main drawback in this work is that the decision making is often slow compared to other models.

Hui Wu et al. [9] developed an aircraft detection model in satellite images that depends on a new aircraft detection framework based on objectiveness techniques BING and CNN.

CNN helps to gather features from the raw data available and can be automatically used in many object detection activities. BING proposes a candidate object region that saves time as well as improves the detection rate. This paper took a dataset from aircraft detection in Google Earth. The limitations of this work are that variations in aircraft type, such as pose and size, decrease the efficiency.

Xueyun Chen et al. [10] proposed a system for detecting aircraft in high-resolution remote sensing images. It helps to reduce the influence of gray threshold images. DBN algorithm is used, which can detect the ships in blurred images. These traditional classifier methods increase accuracy. The main limitation in this work is that small-object detection becomes a bit difficult. Bo Du et al. [11] developed a model on a weakly supervised learning framework. It uses coupled convolution neural networks. High-level feature extraction was done for the objects. The main pitfall in this analysis is the difficulty in extracting high-level features.

Kun Zhao et al. [12] developed a model for small-object detection on remote sensing images. Two CNN models, namely one-stage and two-stage object detectors, were used in image classification for object detection. The main pitfall in this analysis is that one-dimension clusters were not modelled appropriately. Ren Ying et al. [13] developed a model for detecting airplanes and ships around the world through remote sensing images. This paper discusses a model on NVIDIA TX2 and YOLOv3 algorithm. The main drawback in this analysis is that the detection speed is comparatively more when compared to other popular models.

Liming Zhou et al. [14] developed a model on Multi-scale Detection Network (MSDN). It can detect small-scale aircrafts even during background noise. The main drawback in this analysis is that the border prediction and the co-ordinates may not have high accuracy all the times. Gang Tang et al. [15] proposed a methodology named N-YOLO, which consists of Noise Level Classifier (NLC) and SAR-target Potential Area Extraction Module (STPAE) in addition to a YOLO detection module, which helps in building a model that can perform competitively with respect to several CNN algorithms. This method has good performance for ship detection using SAR images. Its applications can be extended to territories of marine monitoring, shore to ship identification, etc. The main limitations of this model is that the ship edge information is damaged because of N-YOLO.

Lichuan Zou et al. [16] proposed a model in which multi-scale Wasserstein Auxiliary Classifier Generative Adversarial Networks (MW-ACGAN) is used along with YOLOv3. This model helps to generate an improved network that can be used to generate high-resolution SAR images. The model shows an accuracy of about 94%, which is more highly effective than YOLO. The pitfalls of this model are that (i) it performs effectively only on small sample size, and (ii) it is effective only under particular sea conditions, incident angles, and polarization modes.

Nie Xin et al. [17] proposed a model in which the network prediction layer includes the prediction box uncertain regression. Negative logarithm likelihood function and improved binary cross entropy functions are together used for redesigning the loss function. K-Means clustering technique is primarily used. The Non-Maximum Suppression (NMS) algorithm is used along with Gaussian Soft Threshold function for prediction of boxes. The only limitation is that when given large sets of input data, the model gives a varying accuracy.

Lena Chang et al. [18] proposed a method in which ships can be managed during day and night. The proposed model uses multi-scale feature extraction, which is extremely helpful in recognizing small targets. The training data set has six different categories of ships, which sums up to 5513 visible and IR images collected from various harbors in northern Taiwan. The proposed model achieved 89.1% map score and 24.3 BFLOPs. The limitation of this model is that its performance is limited when the input images are affected by various climatic conditions.

Yash Chaudhary et al. [19] proposed a model based on YOLOv2. This model proved that the average precision (AP) score of YOLOv3, which was around 90.25, was comparatively greater than that of YOLOv2, which produced around 90.05. Another crucial

takeaway of this model is that the inference time is 22 ms, which is far better than YOLOv2, which had 25 ms. The dataset used consists of Gaofen-3 along with images of Sentinel-1. The drawbacks of this model are that (i) it is adversely affected by the climatic conditions, and (ii) it does not work on videos.

Mingming Zhu et al. [20] proposed a model to address various problems related to detection speed and low accuracy in SAR images. An end-to-end strategy for ship detection using YOLOv3 was proposed. The position of the ship can be directly determined using the dimension clusters. A multi-scale output is obtained, and it consists of high-level semantic information from high- and low-level feature maps. The pitfall of the proposed methodology is that it has low accuracy in case of foggy climates, which denies the sharpness of pixels in input images.

Chun Liu et al. [21] proposed a model by adjusting and optimizing various parameters. Deep learning is used in conjunction with target HSV color histogram features and target's LBP local features. A self-correction network is incorporated to correct any drift and jitter in the training samples. This model depicts robustness and stability and can be applied to various territories, such as statistics extraction and waterways videos. The main limitations of the proposed strategy are that (i) there is a mishap if ships appear to overlap each other, and (ii) there is a need to follow-up research for optimization.

Gang Tang et al. [22] proposed a model based on the pre-selection of region of interest (RoI), which produces a novel high-resolution image network. The model is tested against Google Earth images and HRSC2016 datasets. This model has an improved 16.19% accuracy and 19.01% higher recognition rate. The main takeaway of this model is that the intersection over union (IoU) value is escalated. The disadvantages concerned with the model are that (i) the dataset used is outdated, and (ii) it does not work well with hued images.

Tianwen Zhang et al. [23] proposed a mechanism in which deep learning network with histogram of orientation gradient feature fusion (HOG-ShipCLSNet) is used. Four mechanisms are proposed that improve the classification accuracy. The model is tested on two different SAR ship datasets, and exceptional performance is observed, outperforming other mechanisms. For each of the four mechanisms, effectiveness study and feasibility study are performed. This method is proposed mainly due to the negligence of traditional mature hand-crafted features, and it relies heavily on abstract features by major population of CNN SAR ship classifiers.

Tianwen Zhang et al. [24] proposed a mechanism known as balance scene proposed mechanism. In this work, a generative adversarial network is used for scene feature extraction from SAR images. Later, k-Means algorithm is used for creating a scene binary cluster followed by replication of small clusters and other transformative methods to deduce the scene learning bias to finally obtain greater detection accuracy. The proposed mechanism BSLM is very effective, as it significantly improves performance and accuracy even in case of complex inshore scenes. Furthermore, another added advantage is that the scene learning bias is reduced. The proposed mechanism is simple and at the same time effective.

Tianwen Zhang et al. [25] proposed a model based on SAR Ship Light Weight Detector. This model uses few kernels and layers to ensure the lightweight proposed attribute. This model also discusses the FF Module, FP Module, and also SSFP module for the prediction of accuracy loss. ShipDeNet-20 is formed from scratch and is lightweight when compared to existing works. Moreover, it is also used in real-time detection.

Tianwen Zhang et al. [26] proposed a model to solve the problems that were present in the current SAR ship classifiers. The main problem is that polarization is being used insufficiently. To overcome these defects, a model is proposed called polarization fusion network, which has a geometric embedding feature.

Table 1 represents summary of the literature survey. Different approaches and mechanisms are reviewed, and both advantages and disadvantages are portrayed.

**Table 1.** Summary of Literature Survey.

| Author | Tolerance to Noise | Designed/ Implemented | Method of Implementation | Security to Detected Ship Region | Integrity |
|---|---|---|---|---|---|
| [1] Xi Li | No | Designed | Not Specified | No | No |
| [2] HaoLin | No | Implemented | Enhanced YOLOv3 | No | No |
| [3] WangYu | No | Designed | K-Means | No | No |
| [4] Zhonghua | No | Designed | SVM | No | No |
| [5] Bo LI | Yes | Designed | Segmentation classifier | No | No |
| [6] Xuan Nie | No | Designed | R-CNN | No | No |
| [7] Guang Yan | Yes | Implemented | Candidate Selection | No | No |
| [8] Tonjie.N | No | Designed | Bayesian | No | No |
| [9] Hui Wu | No | Implemented | CNN | No | No |
| Proposed Method | Yes | Implemented | YOLOv3 with SHA-512 | Yes | Yes |

## 3. Proposed Methodology

Figure 2 represents architectural model diagram proposed for Security-based Deep Learning Model to detect ships. The model consists of several steps that are to be procedurally followed to obtain the desired outcome. The dataset on a whole consists of 231,722 images. In the model that is developed, 70% of the data is used for training, and the remaining is considered for testing.

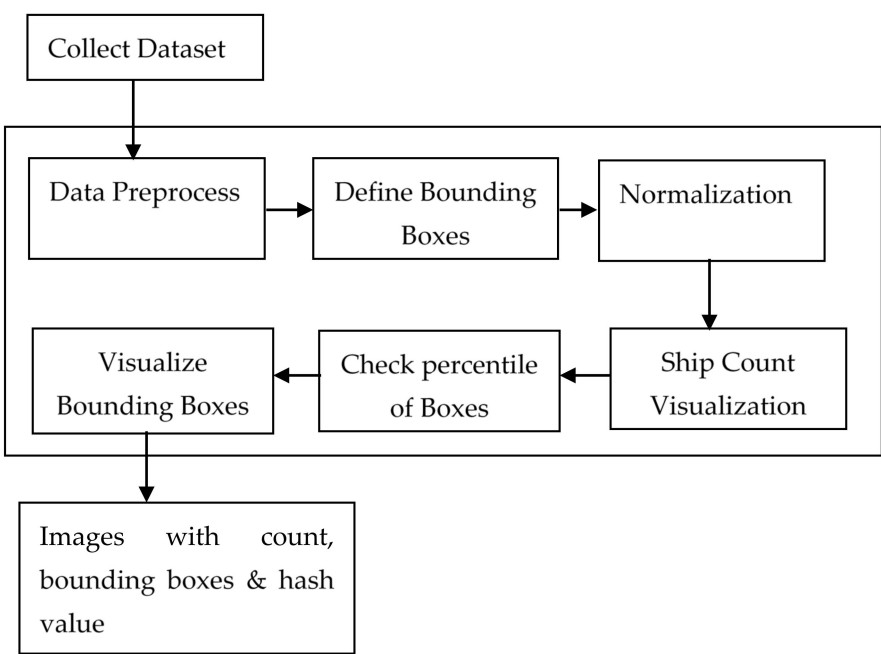

**Figure 2.** Proposed Architecture.

At first, the model is fed with input data followed by finding unique images. As a next step, a definition for the bounding box function is proposed, and then, normalization and dropping encoded pixels based on some criteria takes place. This process is followed by checking the percentile of boxes in each encoded image, and then, downsampling takes place. The data are completely visualized, and then, the security hash is applied to the

images along with the count detection. The next module is the input of training data. The model is tested using the input test data, which is obtained from Kaggle airbus ship detection dataset.

Algorithm 1 depicts how the complete process of ship detection using YOLOv3 takes place. Initially, it fetches data from the dataset, and it finds unique images from the dataset using groupby function and aggregate function. Then, a function to define bounding boxes for the encoded pixels takes place. Then, the data are visualized, and normalization and removing of those with area of Bounding Box less than 2 pixels from the dataset occurs. Then, the hash is applied on the combination of ship count and bounding box location in the image.

---

**Algorithm 1: Ship Detection**

---

Input: Images, D.
Output: Detects the region of ships in image.
Process
Step 1: Read the dataset.
Step 2: Find unique images.
Step 3: Define function to encode bounding boxes.
Step 4: Plot the bounding box areas.
Step 5: Remove boxes that are less than 1 percentile or BoundingBoxArea < 2 *Pixels*
Step 6: Detect the area and show the detected region of ships.

---

Algorithm 2 depicts how the count for the ships is obtained. The training images are given as input to the algorithm, and count is given concatenated with the image as output. At first, kernels are applied, and the downsampling takes place. Then, anchor bounding boxes are applied at each scale in each layer. Then, independent logical classifiers are used to avoid any incorrect classification of ships. Finally, the count is obtained.

---

**Algorithm 2: Ship Count After Object Detection**

---

Input: Images with the region of ship detected.
Output: Counts the number of ships in an image.
Process
Step1: Process individual images from dataset.
Step 2: Initialize count to 0.
Step 3:
While the image contains more than 0 BoundingBoxes:
Increment Count
Step 4: Display count.

---

Each image may consist of multiple ships, and each ship is recognized with the help of a single bounding box. Even if the ships are in closer proximity in the image, the bounding boxes overlap a little and can be identified distinctly.

Algorithm 3 depicts how the hash is evaluated using SHA-256. Initially, padding bits are appended, and then, padding length is appended. Now, the total message length should be of a size that is multiple of 512 bits. Next, eight 64-bit buffers must be initialized for 64 rounds. At last, the compression function is applied. Finally, the hash value is obtained.

| Algorithm 3: Integrity Checking of Detected Object |
| --- |
| **Input**: Dataset with region and number of ships already detected.<br>**Output**: Generates hash value for every image of detected ships.<br>**Process**<br>Step 1: Append padding bits to achieve a message length, which is multiple of 512 bit<br>Step 2: Append padding length of 64 bit at end of the message<br>Original message + Padding bits + Padding Length = M X 512 bits<br>Step 3: Initialize values for eight buffers used for 64 rounds.<br>Step 4: Apply compression function. |

The second module is finding unique images. The model then focuses on obtaining unique images to improve the efficiency and accuracy. This process is carried out by using the group by function. As a next step, the index of the entire data frame is reset as follows:

All the attributes are not needed to be provided, while the inplace attribute is a Boolean value that makes any changes to the original data frame only if it is proposed as True. The third module is function declaration to encode bounding boxes. As a next step in developing the model, a function for detecting bounding boxes using encoded pixels is defined. The function is named as rel2bbox, and it accepts two parameters. It involves the complete mechanism to define a bounding box for any ship detected. In case there is any deviation of the image dimensions, the function raises a value error. This function returns four values. The proposed bounding box function helps in accurately tying the ships and bounding boxes together. The bounding box investigates pixel by pixel, and it is an imaginary rectangle that creates a collision box for the target. It is defined by X and Y coordinates. The bounding box accurately bounds the object using rel2bbox function. In the rel2bbox function, the coordinates of bounding box are obtained in such a way that the length and width obtained from shape function of the predicted ship do not exceed the mentioned coordinates, and if it exceeds, masking is done to over-run. The fourth module indicates Normalize- and Drop-encoded pixels. Once the bounding box function is defined, there is a need to normalize the encoded pixels and find bounding boxes for those normalized-encoded pixels. The next step is to drop those encoded pixels from the dataset for which no bounding box is available. For the purpose of dropping the encoded pixels, drop function is used, which is defined in the Pandas library.

The fifth module removes boxes that are less than 1 percentile. The model finds that there are close to almost 1 percent bounding boxes whose area is less than 1 percentile and even some training samples have 0. All these encoded pixels are removed from the data set, and as a next step, distribution of the ships is evaluated. Based on the visualization, the model predicts that most of the ships in the images are very small, and there are only a few that are likely to be large, and one more inference is that most of the images in the training data contain only one ship in particular. Therefore, it confirms that in the dataset, there is a high data imbalance [27–30] considering the count of number of ships.

The sixth module performs downsampling. Since there is large volume of data, only 1000 images from each class are considered for evaluation purpose. This is done with the help of a lambda function. This is followed by resetting the index of the current data frame. Then, approximately 8000 images are available for training the network. The seventh module includes hashing based on the count of the ships and the position of the bounding box within the image dimensions; a hash value is computed and displayed on top of the output. This would facilitate more security when the images are to be transmitted safely.

## 4. Results

The proposed model identifies the ships from the given satellite images. The model can clearly identify the ships even in cloud masking and land masking regions. Figure 2 shows the outputs identified by the given model. Figure 3 represents the output of model, which shows the ship detection region and the count of ships.

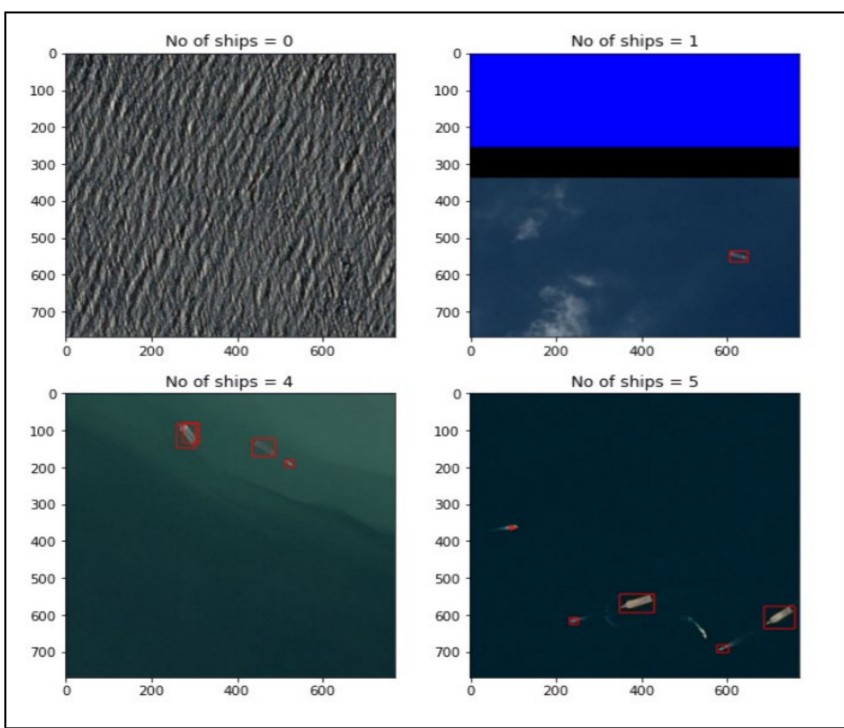

**Figure 3.** Ships identification and count of ships—Case I.

Figure 4 represents the number of ships in particular satellite image and the detection region of ship present. The region is detected and recognized with the help of a red-colored, rectangular bounding box. The count is also identified with the help of bounding boxes present in each image.

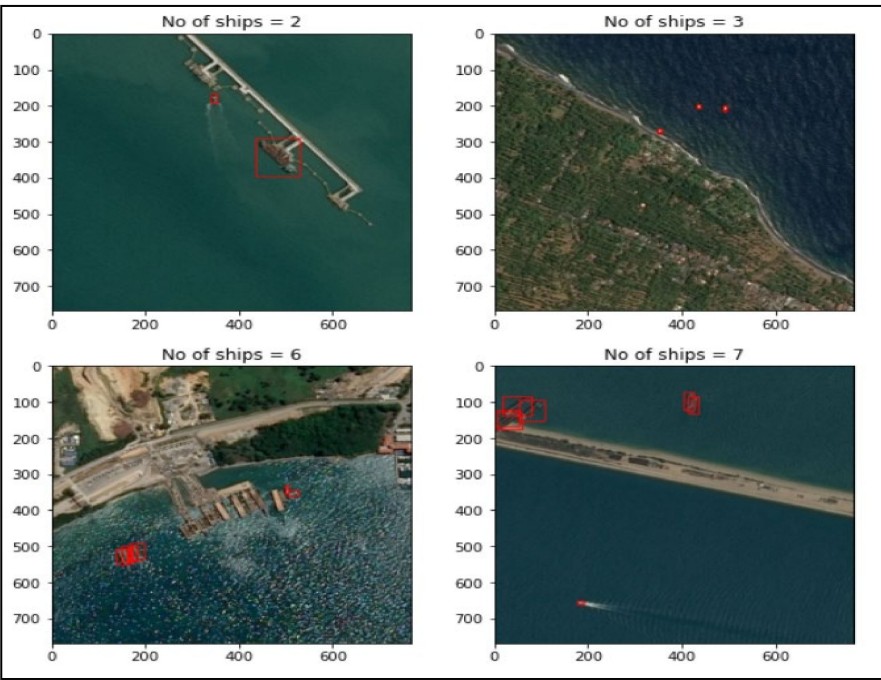

**Figure 4.** Ships identification and count of ships—Case II.

This paper also specifies a security feature that makes the model more secure. For this, the detected area where the ship is present will be given a unique hash value so that, on the other side, if an authorized person wants to know about the presence of a ship, they

can identify with the given hash value. This feature adds security to the model and blocks unauthorized persons from knowing about the presence of ships in the region. Figure 5 represents the output after the hashing algorithm is applied in order to ensure the security.

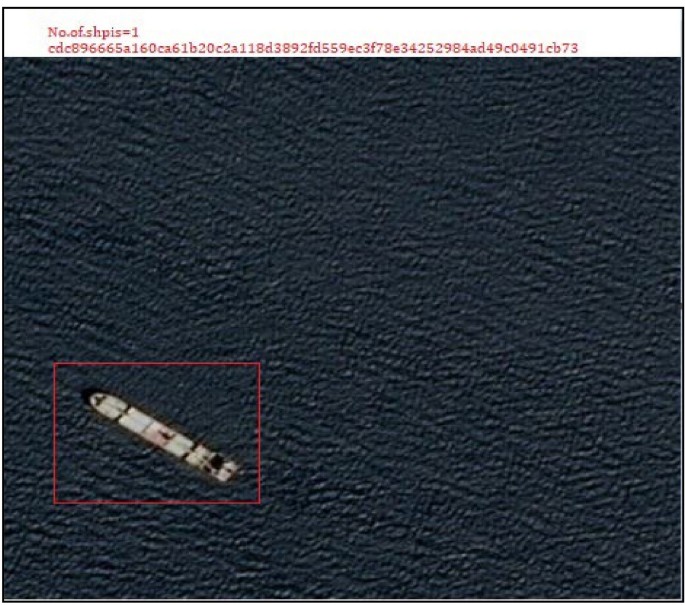

**Figure 5.** Hash value for ship region 1.

Figure 6 represents the count of the ships along with the hash value for the detected region so that only the hash value can be passed to the end-user. For this, SHA 256 is used. It is a cryptographic hash that generates a unique 256-bit (32-byte) value. The hash is completely different from encryption, and it is of fixed length, which cannot be reversible. It is impossible to get back the original value from the hash value generated by SHA-256. This feature makes it more secure. For the brute force approach, it takes 2^256 approaches, which is more than the number of atoms in the universe.

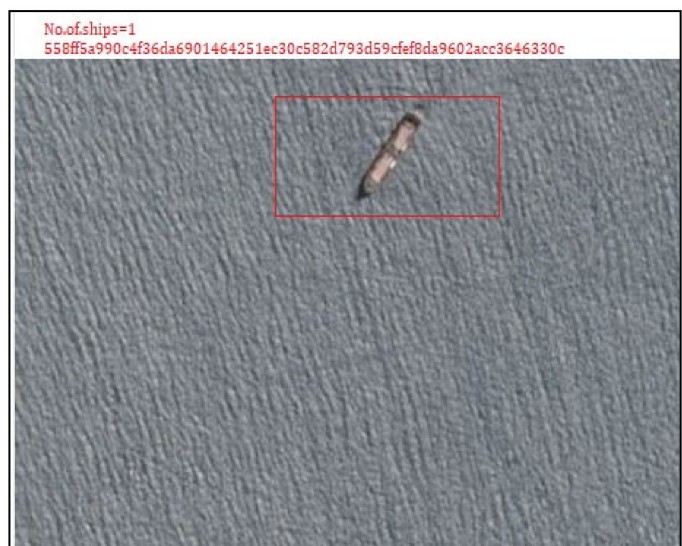

**Figure 6.** Hash value for ship region 2.

In real world, this model accompanied with the concept of hashing can be used to detect lost ships in an army, illegal goods transportation, and for many other purposes. The Synthetic Aperture Radar (SAR) images are initially collected by the satellites and then sent to the concerned authority. If the ship with bounding box region is shared directly during

the transmission, it can be misused by unauthorized parties. In case of highly confidential data, such as naval images [27,28], if one country uncovers other countries' details, then there may be leakage or tampering of national security data. In order to protect the ships from illegal usage and data falling into wrong hands, the hash value of the detected region along with the ship count is transmitted so that only an authorized person can know the presence of ships and their exact location. In such scenarios, the images can be attached with a secure hash value obtained from un-cracked algorithms, such as SHA-256, so that the images remain authentic without any replacement. The concept of hashing is preferred for use with the model since hash is singular and irreversible. If no security mechanisms are used to protect the SAR data, it can become a potential target and can be easily attacked since all the data originate from a satellite and are initially centralized. Therefore, efficient algorithms are used to ensure safe data. Moreover, the model can be applied to other territories, such as tracking merchant ships, finding missing ships, etc.

Table 2 shows the hash values of the recognized ships and their integrity values in terms of hash values. This method can detect the integrity exploitation attacks on satellite data. Moreover, it can be useful if any modifications and alteration happen, such as a man in the middle attack, during the processing time.

**Table 2.** Integrity Check values of the sips region from Case 1.

| | Case I (Figure 5) | |
|---|---|---|
| **No. of Ships** | **Ships Regions** | **Integrity Check Values** |
| 0 | No ship | No Hash Value |
| 1 | Ship 1 | ac4b91dd0de1544d87d20b4561fa2faa7177b5c50aacd06efddd4fdfc29b3a65 |
| 2 | Ship 1 | 7769b5df1b5e26513c8f1e8908a5dd9a928d82fbdbe8009ee2d1707b82bf5383 |
| | Ship 2 | 3a0e1124d1da616cc7c7e9b689e625ad718e9f412bf9bc1b68249453fdcf5be5 |
| 3 | Ship 1 | 4cb429d0d23c96b643d2222eb1c83da69f53a3fa107bd9aed2c8ccc809bdfbaa |
| | Ship 2 | b14918de4acc585f29d0f67f27fceb4edefa37e0ddbd89aff820b5e610e1b57c |
| | Ship 3 | 400564f923d6f37b41ed6e2f3cebdcde407ae791731df17660de66f9810d702c |
| 4 | Ship 1 | eca67246bb6b4d6aaea7bc3cfd6788ef8013c8997c1ea2c0469aba10b12a200b |
| | Ship 2 | 8f0f7eefd0bfe26c6d7b090d74b6a815dcd29ac7b1b5a7a7d92f7a706c771b85 |
| | Ship 3 | 11a8f1bc9cba81923e85b5f123db9379c17ee6758d16ca5ab7dd6bdccf20e040 |
| | Ship 4 | 6f0f9e079cd36d853cafd4980260a09a7248fd20739dc6cabe410b77b3399bae |
| 5 | Ship 1 | 19fe13d266cc988d0540ac9b6e117439958b639ba9e3c2d41338aaa3d53ce2c0 |
| | Ship 2 | 6502de7666ba0f8df3a2fa90b9400bd28d4174b7b8225caa76961aba171ab80e |
| | Ship 3 | 7b33d682e0163f671adbfedcfc9739f082ffd951ca0ea8ac6a92ba7e5e317068 |
| | Ship 4 | b74fd1d48c2288090c712d7580808c7936a8659f90fd7edbdf3053aa5a610676 |
| | Ship 5 | ecdabaa90f4f520862dba2e08ec54568630a9fc9339e023bec8157ae96b6ac8d |
| 6 | Ship 1 | ef159b9a5f0dd433c2b8b51e81b68f2413eb0b8e10183ee6808e120f0ee26dcc |
| | Ship 2 | b532035a537601cc11c17b5d59863856d01ddf912cca2e6cebf077ce4677f350 |
| | Ship 3 | 51bee614135721f668bbf6fbdfdcbed997ff2fa9065fe50fcfe7e91155281289 |
| | Ship 4 | 716d7cf20fa33abe9034bbb4e9ca2213803cd700d3c330fd75a7d4edde181809 |
| | Ship 5 | dbf561608095de77fdbf0531b7740297f8d430517ef6524e0f89543e4d63f4cc |
| | Ship 6 | 2a7c1faef5720df008fda22e4a5e67f916c66a5f9aa9c0e53c5f7c45ae1d966c |
| 7 | Ship 1 | 6faad500283beec08c964adbf2bdb6c3ec8bf3a5f1e7cb2b410ad9280e273818 |
| | Ship 2 | 42354546e839d24a1236618aadfaeeed8feb8f2ad2bf031c96a0bb9da3b0c90c |
| | Ship 3 | 1e503fa00bed77bb469e7d0665cc2de9ad86c9a2fadbafe7ce9ad3a92b303057 |
| | Ship 4 | 12272102b403459e08b543fe5ea8ab4e9567f078a932319861ccbb11ee5b6f2e |
| | Ship 5 | 32ef26a1cbb388c56642f83ae5b21e47ef11ac6bbca06dd30d2739d980f3f93f |
| | Ship 6 | 0c3dc25f5ba3238b4e724da0da4ee1973b06199f6c5ac62b53418fc1e142b3b9 |
| | Ship 7 | af362f1662eb33bac0c7838b95fa9abfb0c32e70ad5b7c054aca84306559ac5c |

### 4.1. Performance Analysis

The dataset is sourced from Kaggle. The dataset is divided into three parts said to be training data, testing data, and verification data. This paper uses a common split where

70% is for training and 30% for testing. Table 3 represents different domains of splitting of dataset. The following were the evaluation indicators used for checking the performance of the model.

**Table 3.** Splitting of the Data.

| Class Name | No. of. Samples |
|---|---|
| Training Data Set | 162,205 |
| Testing Data Set | 69,516 |
| Total | 231,722 |

### 4.1.1. Confusion Matrix

Confusion matrix is a table that is used to check the performance of the model. Confusion matrix is simple. It is used on classification problems to derive summary, and it presents the number of true and false positives and negatives along with count. Table 4 represents the confusion matrix and the accuracy calculated for different predictions by the proposed model. True positives are those where the model interprets as true, and the actual value is also true. Here, the ship is predicted at the correct region. False positives reveal that the model detects at the incorrect position, and the ship that is positioned was not detected. False negatives represent an incorrect prediction of the ship, and finally, the true negatives represent the prediction at invalid positions is true.

**Table 4.** Confusion Matrix.

|  | Actual Values Part of Positive Prediction | Actual Values Part of Negative Prediction |
|---|---|---|
| Values that are predicted as positive | Rate of True Positives (93.4%) | Rate of False Positives (2.3%) |
| Values that are predicted as negative | Rate of False Negatives (4.8%) | Rate of True Negatives (96.8%) |

Rate of true positives describes the proportion of actual ships that were exactly predicted as ships. Rate of false positives describes the proportion of those ships that are not actually ships but were predicted as ships. Rate of false negatives describes the proportion of those ships that are not predicted as ships but were actually ships, whereas rate of true negatives describes the proportion of those ships that are not actually ships and are not predicted as ships.

The proposed paper discusses different metrics to evaluate a model. The performance evaluation metrics were accuracy, recall, precision, and F-measure [31–34]. The model also predicts well under cloud masking, land masking, and even small, blurred images due to the powerful data processing feature of YOLOv3. Multi-level detection helps to achieve the advantages compared to other models. Because YOLOv3 detects and classifies objects at three different levels, those constraints experienced by other methodologies can be outperformed.

### 4.1.2. Accuracy

From all the given classes, accuracy specifies how many images are predicted correctly. If the model shows more accuracy, then the performance of the model will be good. Let true positives be defined as TPO, false positives as FPO, true negatives as TNN, and false negatives as FNN.

$$\text{Accuracy} = (TPO + TNN)/(TPO + TNN + FPO + FNN) \tag{1}$$

The accuracy of the model on substituting above values will be equal to 96.4% It indicates that from 100 samples, the model can absolutely predict 96 samples as correct.

Figure 7 represents the bar graph for different percentages of accuracy at different conditions in the satellite images by considering the proposed approach.

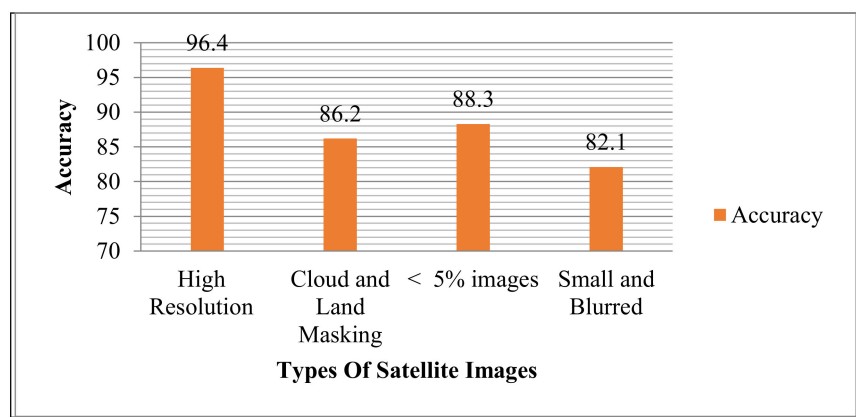

**Figure 7.** Accuracy of detection of ships for different file types.

### 4.1.3. Recall and Precision

Recall is defined as number of samples predicted correctly from all positive classes. Precision is defined as the classes that are classified as positive that are actually positive. If TPO is True Positive, FNN is False Negative, FPO is False Positive

$$\text{Recall} = (\text{TPO})/(\text{TPO} + \text{FNN}) \tag{2}$$

$$\text{Precision} = (\text{TPO})/(\text{TPO} + \text{FPO}) \tag{3}$$

### 4.1.4. F-Measure

It is mainly used to compare two models. F-measure uses both recall and precision. It considers harmonic mean instead of arithmetic mean. Let R = recall, and P = precision; then,

$$\text{F} - \text{Measure} = (2*\text{R} * \text{P})/(\text{R} + \text{P}) \tag{4}$$

Figure 8 shows the performance evaluation of the proposed application by considering the factors of accuracy, recall, precision, and F-measures. Compared to R-CNN and k-Means, the proposed approach has the better performance in detecting the ships from the satellite images.

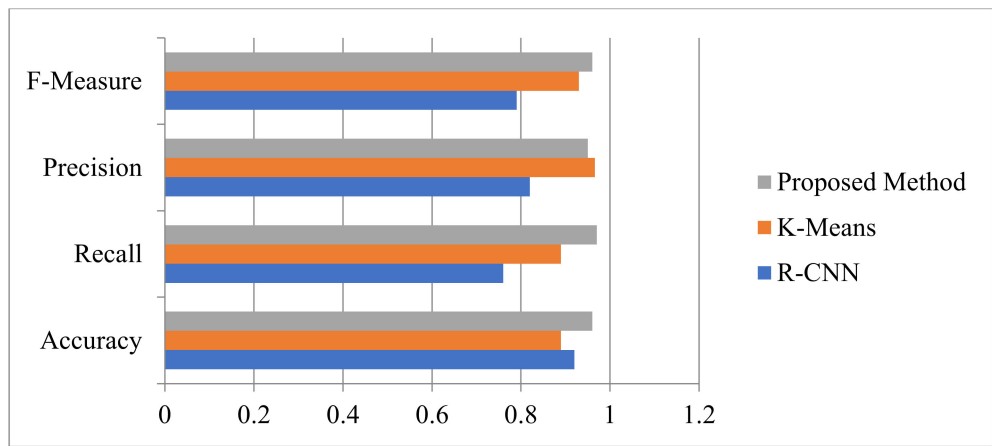

**Figure 8.** Performance comparison with other approaches.

We calculated using three different models, namely YOLOV3, R-CNN, and k-Means. The proposed method using YOLOv3 shows higher accuracy of 96.4%, where R-CNN

shows 92.11%, and k-Means shows 89.8%. Performance analysis using YOLOv3, R-CNN, and k-Means was performed on the same dataset.

## 5. Conclusions

In this paper, a secured, deep learning model for detecting ships is proposed. The developed model is accurate with real-time data and performs well under adverse climatic conditions due to the use of YOLOv3, which performs detection at three different layers. In addition, keen interest is taken on pre-processing the input data, as the quality and quantity of the training data deeply impact the performance and effectiveness of the model. Furthermore, the proposed bounding box function helps in accurately tying the ships and bounding boxes together. To improve the effectiveness and decrease the processing time, normalization and dropping those images having area less than 2 pixels was initiated. To add and push the performance of the model, downsampling of data was performed. By using this approach, more unique and import classes of data can be closely visualized rather than dealing with a plethora of data that reduce the effectiveness and increase processing time. The main advantage of the proposed model is that it performs better in comparison to other models in case of cloud masking, land masking, and small, blurred images. Higher efficiency can be observed in case of adverse climatic conditions. Another advantage is that this model can prove equally well-performing with ships of small size, medium size, and large size that are present in the images fed to the model. In the future, it is proposed to extend the model to perform with video inputs as well.

**Author Contributions:** Conceptualization, S.G. and R.C.; data curation, S.G.; formal analysis, R.C. and A.C.; funding acquisition, C.I.; investigation, C.I.; resources, R.C. and T.R.G.; software, S.G.; supervision, T.R.G.; validation, T.R.G.; visualization, A.C.; writing—original draft, S.G., R.C. and A.C.; writing—review and editing, C.I. and T.R.G. All authors have read and agreed to the published version of the manuscript.

**Funding:** This research received no external funding.

**Data Availability Statement:** The dataset used in this work is available at https://www.kaggle.com/pesssinaluca/ship-detection-visualizations-and-eda/data (accessed on 14 December 2021).

**Conflicts of Interest:** The authors declare no conflict of interest.

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
