# Peer review of "Hash-Based Deep Learning Approach for Remote Sensing Satellite Imagery Detection"

_water, doi:10.3390/w14050707_

Round 1
Reviewer 1 Report
The paper is in part okay but many sections require a major rework before - in terms of both content as well as presentation. Figures are incomplete, there are lots of typos as well as formatting is weird at many places making it hard to read. Please see the attached document for detailed comments.
Author Response
We thank reviewer-1 for providing constructive comments. The responses for the queries are as follows:
- Clarify the use of term Quantized in the Title of the Article
Response:
Thank you for the valuable suggestion. We have updated the title.
- Highlight authors contributions properly to help the readers identify the new research contribution
Response:
Thanks for your valuable suggestion, The new research contribution is more clearly described in the abstract of the article.
- Algorithms should be generic pseudo code and not a summary of jupyter notebook/ python code.
Response:
Thanks for suggesting, all Algorithms were updated to represent pseudo code rather than depicting complete functionality using functions.
- Highlight idea behind the algorithm rather than functions
Response:
Thanks for your feedback, all the functions were replaced with more general pseudo code.
- Highlight the use of Hashing concept and provide a discussion on how these will be useful in real-world scenario
Response:
Thanks for your valuable suggestion, The use of Hashing concept and its use in current scenario is specified under Results Section followed by Figure 6.
- Significance of terms used in confusion matrix
Response:
Thanks for your feedback, The significance is described under Performance Analysis section below Table 4 Confusion Matrix.
- Clarify the claim ‘The model also predicts well under Cloud Masking, Land Masking and even small-blurred images’
Response:
Thanks for suggesting, Explanation to the claim is given under the section 4.1 performance analysis below Table 4 Confusion Matrix.
- Clarify the claim ‘the proposed Bounding Box function helps in accurately tying the ships and bounding boxes together’
Response:
Thanks for your valuable suggestion , explanation to the claim is given in the Methodology section.
- Figures are incomplete
Response:
Thanks for your feedback , Figure 2 proposed Architecture and Figure 1 Generic Diagram are updated.
- Modify Typo errors
Response:
Thanks for your feedback, The complete document is reviewed again and all typographic errors are rectified.
- Update formatting
Response:
Thanks for suggesting, Format is updated.
Reviewer 2 Report
This paper can be accepted with a major decision after considering the following critical comments.
- In the section 2 materials and methods, the authors must add the reviews in detail, e.g., shipdenet-20, balance scene learning mechanism, polarization fusion network with geometric feature embedding, hog-shipclsnet, and selpn-dpff.
- Please clarify your method advantages more clearly.
- IMHO, the Conclusion should be re-written to 1) explicitly describe the essential features/advantages of the proposed method that other methods do not have, 2) describe the limitation(s) of proposed method, and 3) what aspect(s) of the pro-posed method could be further improved, why and how.
Author Response
The authors thank reviewer-2 for the positive and constructive feedback. The responses to the comments are as follows:
- In the section 2 materials and methods, the authors must add the reviews in detail, e.g., shipdenet-20, balance scene learning mechanism, polarization fusion network with geometric feature embedding, hog-shipclsnet, and selpn-dpff.
Response:
Thanks for suggesting. In the section 2 materials and methods, we have added more reviews in detail according to the examples given. Selpn-dpff review was not added as we could not find the exact reference.
- Please clarify your method advantages more clearly.
Response:
Thanks for your feedback, The advantages of the proposed method are elaborated in the conclusion section.
- IMHO, the Conclusion should be re-written to 1) explicitly describe the essential features/advantages of the proposed method that other methods do not have, 2) describe the limitation(s) of proposed method, and 3) what aspect(s) of the pro-posed method could be further improved, why and how.
Response:
Thanks for your feedback, The advantages, limitations and further improvements are clearly mentioned in the conclusion section.
Round 2
Reviewer 1 Report
The manuscript has improved a lot but still required major work - both in terms of presentation as well as formatting (figures). Please see the attached pdf file for more detailed comments. Please highlight how you propose usage of hashing in the system and why this is significant. This is a major contribution of the paper compared to earlier and should be discussed in detail.

Author Response
Responses
Title: Hash based Deep learning Approach for Remote Sensing Satellite Imagery Detection
Manuscript ID: water-1533079
REVIEWER #1
We thank the reviewer for the constructive comments. Our responses to the queries are as follows:
- Highlight the purpose of hashing in the system and why this is significant.
Response:
Thanks for the suggestion. The explanation for use of hashing in the model is clearly mentioned under the results section below Figure 6.
- Repetitive- please rephrase YOLOv3 in abstract.
Response:
Thanks for mentioning. Necessary changes are made and sentences are rephrased. The changes are highlighted in the Abstract.
- Formatting in Figure 1.Generic Diagram
Response:
Thanks for the suggestion. Necessary changes and modifications are made in the figure.
- In Table 1. Summary of Literature Survey, modify typographic errors
Response:
Thanks for highlighting. The typographic errors specified are rectified and the complete paper is reviewed again and all necessary changes are made.
- Formatting in Figure 2. Proposed Architecture can be rectified.
Response:
Thanks for the suggestion. Formatting of the suggested figure is improved.
- Changes in Algorithm 1.
Response:
Thanks for mentioning. As suggested, modifications are done to the algorithm steps removing the descriptive functions used in model and using explaining basic control flow.
- In Algorithm 2, what if a bounding box has more than one ship? Is that not a possibility?
Response:
Thanks for mentioning. The required explanation for the posed question is given under Algorithm 2.
- In Algorithm 2, reformat sentences to not having typographic mistakes.
Response:
Thanks for mentioning. Typographic mistakes are reviewed and changes are made.
- In the Results section, Quality of output images is to be updated.
Response:
Thanks for suggesting. High resolution images are included.
Reviewer 2 Report
thanks the authors' swift corrections. accept as it is.Author Response
The authors thank the reviewer for the positive feedback.
Round 3
Reviewer 1 Report
The paper presentation and quality has improved significantly and looks to be in good shape except for Figure 1 where one can see line numbers appearing inside the figure - this needs to be corrected in the final submitted manuscript. I have no further comments but would advise the authors to do a final proofreading of the paper before submitting the final version.